# Adenovirus Terminal Protein Contains a Bipartite Nuclear Localisation Signal Essential for Its Import into the Nucleus

**DOI:** 10.3390/ijms22073310

**Published:** 2021-03-24

**Authors:** Hareth A. Al-Wassiti, David R. Thomas, Kylie M. Wagstaff, Stewart A. Fabb, David A. Jans, Angus P. Johnston, Colin W. Pouton

**Affiliations:** 1Drug Delivery, Disposition and Dynamics, Monash Institute of Pharmaceutical Sciences, Monash University, Melbourne 3800, Australia; angus.johnston@monash.edu; 2Department of Biochemistry and Molecular Biology, Monash Biomedicine Discovery Institute, Melbourne 3800, Australia; thomas.d@unimelb.edu.au (D.R.T.); kylie.wagstaff@monash.edu (K.M.W.); david.jans@monash.edu (D.A.J.); 3Drug Discovery Biology, Monash Institute of Pharmaceutical Sciences, Monash University, Melbourne 3052, Australia; stewart.fabb@monash.edu

**Keywords:** preterminal protein (pTP), terminal protein (TP), nuclear localisation signal (NLS), adenovirus, DNA binding proteins (DBP), viral genome, DNA viruses

## Abstract

Adenoviruses contain dsDNA covalently linked to a terminal protein (TP) at the 5′end. TP plays a pivotal role in replication and long-lasting infectivity. TP has been reported to contain a nuclear localisation signal (NLS) that facilitates its import into the nucleus. We studied the potential NLS motifs within TP using molecular and cellular biology techniques to identify the motifs needed for optimum nuclear import. We used confocal imaging microscopy to monitor the localisation and nuclear association of GFP fusion proteins. We identified two nuclear localisation signals, PV(R)6VP and MRRRR, that are essential for fully efficient TP nuclear entry in transfected cells. To study TP–host interactions further, we expressed TP in *Escherichia coli* (*E. coli*). Nuclear uptake of purified protein was determined in digitonin-permeabilised cells. The data confirmed that nuclear uptake of TP requires active transport using energy and shuttling factors. This mechanism of nuclear transport was confirmed when expressed TP was microinjected into living cells. Finally, we uncovered the nature of TP binding to host nuclear shuttling proteins, revealing selective binding to Imp β, and a complex of Imp α/β but not Imp α alone. TP translocation to the nucleus could be inhibited using selective inhibitors of importins. Our results show that the bipartite NLS is required for fully efficient TP entry into the nucleus and suggest that this translocation can be carried out by binding to Imp β or Imp α/β. This work forms the biochemical foundation for future work determining the involvement of TP in nuclear delivery of adenovirus DNA.

## 1. Introduction

Adenoviruses are double-stranded DNA viruses that deliver their large 35 kbp genome to the nuclei of nondividing cells. The intracellular trafficking of adenoviruses has been studied extensively, but the mechanism that facilitates the final step of import of adenovirus DNA through the nuclear pore is not well-understood [1]. Unravelling the nuclear transport mechanism at the molecular level will provide insight into the import requirements of large DNA molecules and could provide a valuable blueprint for the design of synthetic pharmaceutical gene delivery systems. Adenoviral DNA is assumed to enter the nucleus as a complex with some or all of the viral core proteins [2,3]. However, which of these components is essential for import is not clear. The core of the adenovirus has three major proteins—pV, pVII and adenovirus terminal protein (TP)—that collectively compact and protect its genome. One molecule of TP is covalently attached to each DNA strand at the 5′ end linking to the phosphate group in the DNA backbone. It is first expressed within a host cell as preterminal protein (pTP), and subsequently translocated to the nucleus where it is coupled to newly replicated viral DNA. pTP is later cleaved by adenovirus proteases to produce TP as each viral particle matures. This unique covalent DNA–protein link is established at pTP Ser–580 (referred to in this study as Ser–562, see experimental procedures below). pTP is critical for adenovirus polymerase priming and subsequent adenovirus replication [4,5]. In the infecting virus, the viral DNA appears to use its attached TPs as nuclear matrix anchors [6], thus enhancing its extrachromosomal stability and improving adenovirus gene expression. Previous work indicated that major core protein pVII, a DNA–condensing protein, appears to play an essential role in DNA delivery [7]. Whether TP assists the incoming viral DNA during nuclear entry is not yet clear.

Studies of adenovirus TP have been limited, and the relationship between adenovirus TP and its nuclear entry is also not well-defined. An early study identified a positively charged motif of pTP that contains a nuclear localisation signal (NLS), PV(R)_6_VP. This motif was reported to be essential for adenovirus pTP translocation into the nucleus by observing codelivery of the associated viral polymerase [8]. The positively charged motif identified is upstream of a proline-rich area within the pTP sequence. The removal of the region containing the PV(R)_6_VP sequence inhibited TP nuclear transport. The study’s authors concluded that the TP NLS resembled the prototypical simian vacuolating virus, (SV40) large T–antigen sequence and was necessary for pTP delivery to the nucleus.

NLSs are drivers of protein transport across the nucleus [9]. Their presence has also been implicated in the delivery of large DNA molecules [10]. NLSs resemble the SV40 T–antigen sequence (PKKKRKV) and typically bind with importin α (Imp α) protein adaptors. Complexed Imp α then binds to importin β (Imp β) through its conserved importin β binding domain (IBB) domain [11] to deliver the protein cargo. Other forms of NLSs may translocate to the nucleus without binding to Imp α. This nuclear delivery can be achieved by directly binding to Imp β [12] or one of its >20 homologues [13]. NLSs can either be monopartite, carrying one essential positively charged region or bipartite, such as the classical nucleoplasmin signal [14], with two separate but interdependent positively charged peptide motifs separated by a 10–12 bp spacer. For optimal nuclear localisation, both regions are required.

Before the role of TP in viral DNA entry can be tested, a detailed understanding of the protein sequences required for nuclear localisation must first be elucidated. We began this study by examining various regions of TP utilising a series of engineered GFP fusion proteins. This strategy allowed us to compare the nuclear localisation of each fusion and led us to identify a key bipartite signal essential for fully efficient nuclear entry of TP. Furthermore, the current study examines the localisation of bacterially expressed TP using biochemical and in vitro assays of nuclear delivery. Finally, in this study, we explore the form of nuclear interaction that led to TP nuclear localisation and identify Imps that could be involved in the nuclear entry of TP.

## 2. Results

### 2.1. TP and Its Precursor (pTP) Localize in the Nucleus

The sequence requirements for nuclear access of TP were initially identified using confocal laser fluorescence microscopy, by analysing the intracellular distribution of GFP fusion proteins (Figure 1), after transfecting cells with mammalian expression plasmids (Figure 2 and Appendix A). All pTP fragments were fused to GFP at their N–terminus (See Appendix A for detailed fragments design). This strategy allowed us to make direct comparisons between the localisation patterns of each fragment in two cell lines—293A cells, a derivative of HEK293 cells with enhanced adherence (from Thermo Fisher), and HeLa cells. Inspection of the pTP sequence suggested three sequences with positively charged amino acids, hereafter referred to as NLS_1_, NLS_2_ and NLS_3_. This hypothesis is based on identifying motifs with rich basic residues that sometimes correlate with the presence of NLS. However, we performed comprehensive fragmentation analysis beyond the hypothesized NLSs. The transfected full-length pTP fusion protein (GFP–pTP) showed strong localisation within the nucleus with some distribution in the cytoplasm.

In contrast, TP–GFP was fully localised to the nucleus (Figure 2A). The nuclear accumulation of both GFP–pTP and GFP–TP in the nucleus was markedly different from the free GFP, which was found in both cell compartments. The ratio of mean fluorescence in the nucleus and cytoplasm (N_f_/C_f_) for GFP was 0.83 compared to GFP–TP with a N_f_/C_f_ of 27.12 and GFP–pTP with a N_f_/C_f_ of 4.91 (Figure 2A and Figure 3B). Statistical analysis showed that in HeLa cells N_f_/C_f_ of GFP was significantly different to that of GFP–TP (*p*–value < 0.0001) but not GFP–pTP (see also Appendix A). The presence of a negatively charged domain downstream of NLS_2_ in the absence of the C–terminal domain of TP (Figure 1) impeded, to a small extent, GFP–F1 localisation to the nucleus, especially in 293A cells (Appendix A), despite the relatively small size of F1 and the presence of the proposed nuclear localisation signal. Nuclear accumulation was enhanced when the negatively charged domain was removed as in F2 (Figure 2A and Appendix A). In Hela cells, the N_f_/C_f_ of GFP–F2 (F2 N_f_/C_f_ = 14.18) was higher than GFP–F1 F1 N_f_/C_f_ = 1.72 and *p* = 0.0413) but still significantly lower than GFP–TP (TP N_f_/C_f_ 27.12, *p* = 0.0055) (Figure 2B).

### 2.2. The Requirement of a Bipartite NLS for Nuclear Localisation of TP

The second and the first NLSs from fragment F2 were further removed to generate F3 and F4 fragments, respectively (Figure 1). The removal of the second NLS blocked nuclear localisation of TP and GFP–F3 was fully localised to the cytoplasmic compartment. A similar pattern of localisation was observed for GFP–F4, which included neither of the putative NLSs (Figure 2A). The mean N_f_/C_f_ values for GFP–F4 and GFP–F3 were comparable to GFP–F1 (*p*–value > 0.9999) but GFP–F3, for example, was significantly lower than GFP–F2 or GFP–TP with *p* values of 0.028 and < 0.0001, respectively (see Appendix A for statistical analysis).

We then generated an additional subset of fragments where the precursor region pTP was removed. In this subset, F5 encoded both NLS_2_ and NLS_3_; F6, encoded NLS_3_ but not NLS_1_ or NLS_2_, beginning from Serine–562; F7 lacked all the potential NLSs but incorporated the negatively charged fragment at its N–terminus; F8 had a similar sequence to F5 but lacked the NLS_3_ (Figure 1). In this construct subset, the loss of either NLS_1_ or NLS_2_, but not NLS_3_, similarly blocked the nuclear localisation (Figure 2). The removal of NLS_1_ impeded nuclear localisation of GFP–F5 and GFP–F8 (Figure 2A) despite the presence of NLS_2_ in both of these fusions, which is the sequence PV(R)_6_VP that was previously proposed to be solely responsible for the nuclear localisation of TP [8]. In this subset, the absence of either NLS_1_ or NLS_1_/_2_ resulted in cytoplasmic accumulation. Specifically, N_f_/C_f_ values of F5–F8 fusions were significantly lower than GFP–TP (*p*–value < 0.0001 compared against F5), and only modestly, but significantly, lower than GFP–F2 (*p* = 0.01 compared against F5), which encoded both NLSs. GFP–F9 and GFP–F10, which both included NLS_1_ and NLS_2_ but lacked NLS_3_ (Figure 1), showed prominent and exclusive localisation in the nucleus (Figure 2A). This localisation was evident in both cell lines and was significantly different from fusions of F3–F8 (Figure 2B), highlighting the importance of both NLS_1_ and NLS_2_ in the nuclear localisation of TP.

The GFP–TP fragment was re-engineered to exclude the possibility that the fragmentation process could have altered the structure in a way that indirectly impeded the nuclear localisation. We used PCR–derived directed mutagenesis to engineer three mutants (Mut_1_, Mut_2_ and Mut_3_) and deletion fragments. These mutants incorporated amino acid substitutions into the positively charged amino acid residues of NLS_1_, NLS_2_ and NLS_3_, respectively. Mutation of NLS_1_ or NLS_2_ (Mut_1_ and Mut_2_, Figure 1) disrupted the nuclear exclusivity of the original GFP–TP. Mut_1_ affected the nuclear localisation of GFP–TP more prominently than Mut_2,_ (Figure 3A,B). The analysis of mean N_f_/C_f_ between GFP–Mut_1_ and GFP–Mut_2_ suggested that the mutants were not significantly different (*p* = 0.9998 and 0.9948 for HeLa and 293A cells, respectively (Figure 3C)). Mutation of NLS_3_ (Mut_3_) did not affect nuclear localisation and Mut_3_ showed a similar N_f_/C_f_ profile to GFP–TP (Figure 3A–C). Finally, NLS_2_ was deleted from the GFP–TP sequence without altering the downstream sequence to generate the Del_1_ fragment (Figure 1). The deletion of NLS_2_ resulted in exclusive compartmentalisation of GFP–TP (Del1) within the cytoplasm (Figure 3A,B). The difference among Mut_1_, Mut_2_ and Del_1_ was not significant (Figure 3C and see also Appendix A). *p*–values between Mut_1_ and Del_1_ were >0.9999 for both cell lines. These results strongly support the hypothesis that both NLS_1_ and NLS_2_ are required for optimum nuclear entry of pTP and TP.

Identification of the NLS sequences allowed us to compare them to available protein sequences. We aligned sequences after using Uniprot Blast service [15]. The alignment shows significant homology between both NLS_1_ and NLS_2_ in multiple adenovirus subtype (Figure 4, highlighted in green). The NLS_1_ also shares a striking homology with importin α1 region (Figure 4, highlighted in yellow), which is responsible for binding to importin β and known as the importin β binding domain (or IBB).

### 2.3. TP Can Be Successfully Refolded after Expression and Extraction from Inclusion Bodies

To study the biochemistry of TP in more detail, we expressed TP in *E. coli*. Protein expression was confined predominantly to inclusion bodies despite numerous attempts to avoid inclusion bodies protein expression. The TP construct produced in *E. Coli* contained a series of fusion tags including a TEV cleavage site. We observed that TP stability was severely compromised by TEV cleavage despite our attempts to alter conditions tested and in all cases—TP precipitated rapidly (data not shown). For this reason, we continued our study using the TP–Trx fusion protein without cleaving with TEV protease. Protease inhibitor cocktail was also added from the point of TP refolding to improve stability of the protein (Appendix A). Figure 5A–C detail the expression of the terminal protein which is described in more detail in the Methods section below.

### 2.4. Bacterially Expressed TP Requires Energy and Cytosolic Factors for Active Transport and Can Be Observed to Translocate to the Nucleus after Cytoplasmic Microinjection

Following expression, we tested the nuclear localisation of TP. We first labelled TP–Trx with Maleimide–AF594 (Appendix A), which labelled both TP and the thioredoxin tag when tested by TEV cleavage (Appendix A). Importantly, TP–Trx remained stable after dye conjugation. The standard digitonin permeabilisation assay (DPA) indicated a robust localisation of TP in the nucleus after 30 min of incubation at 30 °C (Figure 6A). The localisation was dependant on both cytosolic factors (+Cyt) and energy (+E) (Figure 6A). Rabbit reticulocyte lysate (RRL) was used to replace the cytosolic factors in this assay. An energy regeneration system composed of ATP, GTP, creatine phosphate and creatine phosphokinase was included in the import medium (see Methods for description of the import medium). The absence of either energy or RRL led to a significant decrease in the level of TP nuclear accumulation (Figure 6B).

Additionally, the nuclear localisation of TP was inhibited when the cells were treated with wheat germ agglutinin (WGA), which is known to block the nuclear pore complex [16]. The N_f_/C_f_ was 0.82 with WGA treatment, indicating a nuclear exclusion of fluorescence signal, despite the presence of cytosol and energy during the incubation (Figure 6B). The N_f_/C_f_ of both +Cyt − E and −Cyt + E treatments were 1.2 and 1.1, respectively, whereas the N_f_/C_f_ of the complete reaction (+Cyt + E) was 2.84, which was significantly higher (Figure 6B) compared to the other conditions (*p*–value < 0.0001).

To study the nuclear uptake of bacterially expressed TP in living cells, cellular microinjection was used. A total of 20 min after microinjecting with TP–Trx or BSA–Texas Red (BSA–TR), cells were labelled with nuclear counterstain. The microinjection mixture also contained BSA–FITC. Both BSA conjugates are relatively large proteins that cannot access the nucleus without a nuclear localisation signal. The inclusion of BSA–FITC to the mixture facilitated the identification of the injected cells and confirmed the cytoplasmic localisation of the microinjection. This ensured consistency and avoided the inclusion of false positives (accidental nuclear injection of TP) in the analysis. TP–AF594 rapidly accumulated in the nucleus after cytoplasmic injection, while BSA–TR primarily remained in the cytoplasm (Figure 6C). Analysis of N_f_/C_f_ ratio showed significant accumulation of TP–AF594 in the nucleus compared to BSA–TR (Figure 6D). TP localised primarily in the nucleus 20 min after cytoplasmic microinjection, but there was still noticeable cytoplasmic fluorescence (Figure 6C). We observed cytoplasmic TP up to one hour after microinjection, although the nuclear localisation was slightly lower than at 20 min (data not shown). The DPA and microinjection results demonstrate that bacterially expressed TP–Trx translocates into the nucleus and that this process is an active process requiring energy and cellular proteins and factors.

### 2.5. TP Strongly Binds Imp β1 and Imp α/β1

Based on the similarity of the TP NLS_1_ with the IBB NLS as described above, and given our results showing the requirements of shuttling cytosolic factors for TP translocation, we investigated whether TP can interact with either Impβ1 or the Impα/β1 heterodimer using an established importin-binding AlphaScreen assay (see Methods section). TP was incubated with increasing concentrations of Imps or a GST control. To ensure that binding to the Imp α/β1 heterodimer was not merely an association with free Imp β1 or through Impα displacement, GST–cleaved Imp β1 was used, which would be unable to produce a signal upon binding to TP. Neither GST alone nor Impα was able to bind TP (Figure 7A). In contrast, both Imp β1 and the Imp α/β1 heterodimer bound TP with K_d_s of 21 and 59 nM, respectively, with maximal binding to Imp β1 approximately 20% greater than to Imp α/β1 (Figure 7B). This strong binding suggests that TP can utilise either Imp β1 or Imp α/β1 for transport into the nucleus.

### 2.6. Inhibitors of Imp α/β1 and Imp β1 Lower Nuclear Accumulation of TP

The binding data between TP and importins was supported by a localisation study of bacterially expressed TP in living cells. Bacterially expressed and labelled TP (see Methods) was microinjected into Hela cells containing ivermectin, a well-known Imp α/β inhibitor [17] or importazole, a selective Imp β inhibitor [18]. The drugs ivermectin and importazole both inhibited the nuclear distribution of TP (Figure 7C). The mean N_f_/C_f_ of TP (DMSO addition) was 1.62. Incubation of cells with ivermectin lowered the nuclear localisation expressed as N_f_/C_f_ to 0.86 (Figure 7D). Importazole had the same effect, lowering the N_f_/C_f_ to 1.04 (Figure 7D). When the data were analysed statistically (Appendix A), the N_f_/C_f_ values obtained in the presence of the inhibitors were significantly different from the DMSO control (TP vs. TP + Iver with *p*–value < 0.0001 and TP vs. TP + Impz *p*–value = 0.0002). These results show that TP localisation in the nucleus was reduced by including either Imp α/β or Imp β inhibitors. This indicates that TP relies on either Imp β or Imp α/β for nuclear translocation. As a control, a BSA–FITC–NLS conjugate was used, which contains the sequence of T–ag_111−135_. Ivermectin at the same concentration also blocked BSA–FITC–NLS localisation (Appendix A). Leptomycin B, an exportin 1 blocker [19,20], did not affect TP nuclear accumulation measured by N_f_/C_f_ (Figure 7C,D, Appendix A) when used at a concentration of 10 ng/µL (TP + Impz *p*–value = 0.2024).

## 3. Discussion

In this study, we applied bioengineering and biochemical methods to study adenovirus terminal protein nuclear localisation and the mechanism of this localisation. Although adenoviruses are a well-characterised DNA virus model, little attention was placed on studying the intracellular trafficking of adenoviral TP. Sequence analysis of TP led us to propose three potential NLSs (described in this manuscript as NLS_1_, NLS_2_ and NLS_3_).

In the first part of this study, we found that TP and its precursor (pTP) localise in the nucleus. This localisation was independent of the viral polymerase [8], despite the cell-type used. In the first bioengineered fragment, which exhibited inhibition of nuclear localisation, a motif rich in negatively charged residues (EEEEGEALMEEEIEEEEE) remained downstream of NLS_2_. Although the negatively charged domain is present in full-length TP, it did not impact on nuclear translocation as long as it was not the last motif in the C–terminal flank of the protein (see, e.g., F9 and F10). It is possible that the negatively charged domain interfered with the function of NLS_2_ when it lacked a structural constraint provided by the full downstream structure at the C–terminus of the motif. The second part of the study identified the NLS required for nuclear entry of TP. Further fragmentation and mutation studies indicated the requirement of a bipartite NLSs for nuclear localisation of transfected TP fragments. The use of mutations and NLS selective deletion validated that a bipartite sequence, observed in the fragmentation study (F1–F10), was necessary for full NLS activity, not just the result of the structural constraints that result from the removal of motifs from the TP sequence. The data obtained from comprehensively mutating and fragmenting TP indicate that the protein makes use of a more complicated mechanism than the monopartite nuclear translocation mechanism described previously [8].

The difference between our results and the previous study have been revealed by the specific fragmentation approach used in this study. In the previous study, the fragment that was removed by the researchers was a much longer sequence from the TP open reading frame (ORF), resulting in deletion of the accompanying NLS_1_. Interestingly, the NLS_1_ sequence (Figure 4) shows a striking similarity to the IBB domain (an amino acid domain part of Imp α adaptor) in that it facilitates the binding of Imp α to Imp β [11]. Bipartite NLSs have been reported to bind to the Imp α/β heterodimer [21,22]. Alternatively, a nonclassical NLS has been identified which binds directly to Imp β, such as the NLS found in parathyroid hormone-related protein (PTHrP). However, the latter protein is not considered to include a prototypical bipartite NLS with two positively charged amino acid patches [12]. TP has an NLS that is similar to the Imp α IBB domain (Figure 4), and we show in this study that this signal is essential for nuclear entry (Figure 3A–C). The IBB domain found in Imp α binds to Imp β to form a high-affinity α/β heterodimer [23] before shuttling to the nucleus. The idea that the NLS of TP represents a class of bipartite signals capable of directly interacting with Imp β is interesting since this form of binding is not typical of bipartite NLSs. Whether the bipartite signal within TP can bind to Imp β warranted us performing further investigations (see below).

After successful autologous expression, we found that bacterially expressed TP–Trx required active transport to reach the nucleus. We used the digitonin permeabilisation assay (DPA), a widely used assay to investigate the nuclear localisation of proteins. In this assay, we incorporated an anti–DNA antibody which was essential to avoid false-positives due to possible nuclear membrane damage by digitonin treatment. The requirement of energy and cytosolic factors suggests that the TP–Trx can enter the nucleus by interaction with shuttling proteins and the shuttling requires energy to be completed. The nuclear localisation of TP–Trx–AF594 after cytoplasmic microinjection in living cells supports the active transport hypothesis. While this access is robust and statistically significant (Figure 6D), some signal in the cytoplasm remained. An NES signal on TP which may have allowed some continued export of the protein could explain this observation. However, our results following leptomycin B treatment, an inhibitor of exportin 1, showed no impact on nuclear entry of TP compared to untreated control (Figure 7B). The combined results of DPA and microinjection clearly indicate that access to the nucleus from the cytoplasm is an active process requiring energy.

Since active transport requires interaction with a nuclear shuttling factor, we considered which of the cellular factors could be utilised by TP. Our observations that TP can strongly bind Imp β1 and Imp α/β1 were consistent in both the in vitro AlphaScreen assay (Figure 7A) and microinjection in living cells. In living cells, inhibitors of both Imp α/β1 and Imp β1 significantly lowered nuclear accumulation of TP (Figure 7B,D). Although our results strongly indicate that TP can bind and use Imp α/β1 and Imp β1 for nuclear transport, we cannot exclude other cytosolic factors that could assist TP in its nuclear accumulation. Using inhibitors such as ivermectin and importazole, we show significant but incomplete inhibition. We could not use a higher concentration of these drugs as we noticed significant toxicity when attempting microinjection in the presence of higher concentrations of these compounds. As expected, we did not observe strong binding of TP to Imp α in the AlphaScreen assay. This can be explained by the well-known autoinhibition of the Imp α binding site for NLSs in the absence of Imp β [24].

Our study offers an opportunity to study antiviral interventions. Indeed, TP entry to the nucleus is crucial for viral replication and infection [5]. Our study also highlights the possibility of TP involvement in incoming viral DNA entry. Intervening with TP entry to the nucleus can be useful to mitigate adenoviral infections. Adenoviruses are common viruses with respiratory, neurological, eye and gastrointestinal infections [25]. In our study, the use of ivermectin and importazol helped reduce TP entry to the nucleus. Both these drugs and others can be tested on adenoviruses, as can the impact of drugs on viral entry and replication. Our work shows intimate and high-affinity binding between Importin αβ or Importin β. However, structural studies can be used to probe whether TP–importins interaction is unique or whether it can be disrupted by other drugs that are known to affect importin trafficking or interaction.

The results from this work provide an essential foundation to answer whether TP is needed for nuclear entry of adenovirus in the future. The current study focuses on identifying and studying the nuclear localisation sequences of TP and its biochemical interactions with host importins. This aim provides a basic and comprehensive understanding of the mechanism of entry of this protein, prior to studying the TP in the context of viral entry. Future work involving the study of viral particles with mutant TP will be important to clarify the rule of TP in viral entry. However, studying adenoviral TP in the context of nuclear pore entry cycle may pose certain technical challenges. Nuclear localisation of newly expressed TP is a vital first step of adenovirus replication and generating viral particles. Mutations to prevent entry can stop the generation of reporter viral particles for studying the phenomenon of viral nuclear entry. Therefore, development of techniques that circumvent such challenge will be needed.

Interestingly, in some bacteriophage and *Streptomyces* spp. the relevant terminal proteins possess NLSs [26,27] that can enhance gene expression when transfected in mammalian cells. It is yet unclear how the terminal proteins drive nuclear translocation in eukaryotic cells and whether they interact with mammalian nuclear Imps during viral nuclear delivery. This will be an interesting topic of investigation in the future. Our work proposes a new question: is TP involved in adenoviral trafficking to the nucleus? Intracellular viral trafficking is thought to be carried out by a hexon, which the virus uses to dock into the nuclear pore [2]. However, this docking may be incomplete [1]. Previous work showed some involvement of the condensing protein, pVII, in viral delivery across the nucleus [7]. However, in that study, TP was also present in the incoming viral DNA. Tp is present only as two protein copies per viral particle, which makes its identification and localisation difficult. Recently, a biotechnological approach to use TP–DNA as a cloning vector helped produce higher titre of adenovirus after transfection into permissive cell lines [28].

In this study, we used cellular and molecular biology approaches to study TP protein localisation to the nucleus, an essential step in the initiation of adenovirus replication inside the nucleus. Our results show that TP contains a bipartite NLS. Both NLSs must be present for nuclear entry. This entry requires shuttling factors and energy. Finally, the use of Imp β and Imp α/ β heterodimer confirms that potentially either route of nuclear entry is possible. Our data add to the previous description of the TP NLS, explore TP interaction with cellular factors, and form a basis to study the importance of TP in adenovirus infection and entry.

## 4. Materials and Methods

### 4.1. Cloning of Proteins and Fragments

The cDNAs encoding pTP, TP and other fragments were obtained from the Virapower adenovirus plasmid which was used as a template (Thermofisher, Scoresby, VIC, Australia). Fragments of the terminal protein DNA sequence were amplified and cloned using Phusion PCR master mix (NEB, Ipswich, MA, USA) and the oligonucleotides described in Appendix A. This was followed by fragment ligation to the C–terminal end of *Pontellina plumata* GFP using the pMaxGFP vector (Lonza, Basel, Switzerland). The vector was first amplified with PCR (Appendix A—pMaxF, pMaxR). The linear DNA was then gel-purified before assembly. The fragments were inserted into the linear pMaxGFP using a Gibson assembly Kit (NEB, Ipswich, MA, USA), as shown in Figure 1. Oligonucleotides were designed with the assistance of NEBbuilder website and modified using Snapgene software (v. 4.2.4, GSL Biotech; available at https://www.snapgene.com/ ) to ensure at least 20 base-homology with the linearised pMaxGFP PCR product. All Gibson assembly reactions were carried out as recommended by the kit manufacturer, and the amplification products were verified by gel electrophoresis. After fragment assembly into the linearised pMaxGFP, the products of each reaction were transformed into DH5α *E. coli* (NEB, Ipswich, MA, USA) following the manufacturer’s protocol. After streaking, growth and amplification, plasmids from the bacterial clones were purified using GeneJet plasmid miniprep (Thermofisher, Scoresby, VIC, Australia ). For mutations and deletion subfragments, the GFP–TP fragment clone was used as a template followed by PCR amplification to introduce mutations or deletions using the oligonucleotides described in Appendix A (Mut1F/R for mutant 1, Mut2F/R for mutant 2, Mut3F/R for mutant 3 and Del1F/R for the deletion clone as shown in Figure 1). After PCR, products were phosphorylated and ligated using T4 polynucleotide kinase and ligase (NEB, Ipswich, MA, USA) followed by transformation, selection, growth and purification as above to generate the mutated/deleted fragments. The sequence, starting at MEHFLP, was used as an alternative initiation sequence (Gene bank accession AAA92208.1). This placed Serine–580 at the alternative site of 562 (Ser562) when starting from the alternative initiation sequence. No further changes were applied.

### 4.2. Validation of Inserts

All fragments and mutants (Mut1, Mut2, and Del1) were sequenced downstream of the GFP C–terminus to confirm insertion of the fragments and lack of frameshift using Frag–SeqF oligo (Appendix A). Mut3 was sequenced using Frag–SeqFV2 oligo (Appendix A). Sanger sequencing was used to validate all the constructs (performed by AGFR, Melbourne, VIC, Australia).

### 4.3. Mammalian Expression

HeLa and 293A (Thermo Fisher, Scoresby, VIC, Australia) cells were maintained in DMEM + 10%FBS (high-glucose, pyruvate) with antibiotics. Cells were tested for mycoplasma upon their arrival and before creating frozen stocks and were free from contamination. Cells were plated onto cell culture treated µ–slide (ibidi, Martinsried, Planegg, Germany) and using 2 × 10^2^ cells 24 h before transfection. On the following day, purified plasmids were transfected using Lipofectamine 3000/P3000 reagent (Thermofisher, Scoresby, VIC, Australia) following the recommended procedure by the manufacturer. After washing the cells, the culture medium was replaced with fresh growth medium 24 h after transfection. Forty-eight hours after transfections, the growth medium was removed and washed once with Phenol red-free L15 medium (Thermofisher, Scoresby, VIC, Australia), to facilitate growth without CO_2_ in the microscope chamber. This was followed by incubation with L15 medium containing 0.1 mg/mL Hoechst 33342 (Thermofisher, Scoresby, VIC, Australia) for 10 min at 37 °C. In some cases, CellMask DeepRed (Thermofisher, Scoresby, VIC, Australia) was also added to the medium to facilitate cell membrane detection and calculations. The incubation medium was then replaced again with fresh and warm L15, and cells were imaged using confocal laser microscopy with a Leica SP8 equipped with an HC PL APO CS2 63x/1.4 oil-immersion objective under 37 °C. The same settings were applied during image acquisition using LASX software (Leica, Wetzlar, Germany).

### 4.4. TP Cloning and Protein Expression and Purification

The ORF of TP with the TEV cleavage site close to the n–terminus was compiled using gene synthesis. In the construct, Ser-562 was replaced with cysteine residue and cloned in pET32a expression plasmid (Novagen, Darmstadt, Germany) between BamHI and EcoRI sites using directional cloning with restriction enzymes. pET32a sequencing confirmed the insertion of TP. The pET32a–TP construct was then transformed into BL21 *E. coli* (NEB, Ipswich, MA, USA), plated in selection LB media, and then a colony was selected for protein production. After overnight incubation in 4 × 10 mls LB culture, each culture was added to 500 mL terrific or LB broth media and incubated with orbital mixer at 37 °C, 180 rpm. The culture was monitored and then induced for 3 h with a final concentration of 1 mM Isopropyl β–D–1–thiogalactopyranoside (*IPTG*) at O.D._600_ of 1.0–1.2. The bacteria were pelleted and then resuspended in cold lysis buffer (50 mM Tris–HCl, EDTA 10 mM, Triton–X 1% and 5% glycerol + 200 µg/mL lysozyme) and stirred using a magnetic stirrer for 20 min. The suspension was then sonicated (1 min × 3 times using 11 Wattage). Following sonication, DNAse (10 µg/mL) and MgCl_2_ (2 mM) were added to the suspension and incubated further for 20 min while stirring. The suspension was then spun at 10,000 g for 15 min, and the protein expression of TP was identified in the inclusion bodies. The inclusion body pellet was washed once with TritonX containing wash buffer (Tris–HCl 50 mM, TritonX 1%, 1M Urea) and with the same wash buffer lacking Triton–X twice and spun as above after each wash. A total of 30 mL of 6 M GuHCl in TrisAcetate pH 8.6 was used to resuspend the pellet and was incubated overnight at 4 °C to solubilise the protein.

We purified TP from inclusion bodies using FPLC under denaturing conditions (Figure 5A). This yielded high-purity TP (~91%; Figure 5B). To obtain a high yield in a refolded form, we designed an in-house protein refolding screening assay. We selected the optimum buffer system from a group that yielded the lowest aggregation (Appendix A). A buffer composed of 20 mM TrisAcetate, 5% glycerol, 50 mM arginine, and 50 mM glutamic acid at pH 8.4–8.6 was selected.

The TEV cleavage site was accessible, and TP was predominantly cleaved from the fusion construct after 3 h treatment with TEV enzyme (Figure 5C). We were able to detect this cleavage by Western blotting using an antibody against the thioredoxin tag (Trx) (Figure 5C). A band was evident above the cleaved Trx tag, but this band was barely observable with SDS–PAGE (Figure 5C).

### 4.5. TP Purification and Solubility Assessment

TP–Trx containing the His-tag was purified using Histrap 5 mL column (GE healthcare). After binding, column buffer was replaced to wash buffer (20 mM TrisAcetate, 8M Urea pH 8.3) and followed by 3 column washes using the same buffer. After washing, the protein was eluted with wash buffer + 400 mM imidazole. Elution was conducted in a gradient fashion, and the protein peak was pooled.

The assessment of the refolding was performed by testing ~50 different conditions (Appendix A), the original buffers and the additives were made in 4X concentrations initially and then diluted to 1x working concentration to contain the final concentration described in Appendix A. A dialysis membrane of 3K was used to dialyse the TP and was lodged in an Eppendorf tube cap. The tube was cut off at the neck. The sample (50 µL of TP in urea containing buffer) was assembled inside a 50 mL tube containing 50 mL of each different dialysis buffer. The dialyses buffers were replaced once. Following overnight dialysis, the protein solutions were collected and analysed with microplate nephelometer (BMG Labtech, Offenburg, Germany) to determine the extent of aggregation and precipitation.

After determining the best dialysis buffer, the protein was then dialysed using Slide–A–Lyzer G2, MWCO 3.5KDa (Thermofisher, Scoresby, VIC, Australia) and refolded at 4 °C against 2 L dialysis buffer (20 mM TrisAcetate, 5% glycerol, 50 mM arginine, 50 mM glutamic acid pH 8.3). The dialysis buffer was replaced after 6 h, and another 2 L of the same buffer was added, and the unit remained in dialysis overnight. Next day, the protein was concentrated, aliquoted and stored at −80. TEV enzyme was expressed and purified in-house and was a gift from Professor Martin Scanlon at Monash Institute of Pharmaceutical Sciences.

To purify the thioredoxin tag to use as a control, we used TEV enzyme to cleave the TP–Trx. After, filtration was carried out with a 0.22 µM filter. Trx was separated from the other components with using FPLC SEC 70 column (Bio-Rad, Glandesville, NSW, Australia).

### 4.6. Western Blotting

Proteins (TP alone or TEV treated) were run in 12–4% SDS page in parallel. Half of the gel was analysed by Coomassie Blue staining, and the other half of the gel was moved onto a PVDF membrane and sandwiched inside two thick filter papers and run in Trans-Blot SD semidry electrophoresis cell (Bio-Rad, Glandesville, NSW, Australia) for 30 min at 12 V. The PVDF membrane was blocked by Odyssey^®^ Blocking Buffer (Licor, Lincoln, NE, U.S.A.) and labelled with primary mouse Trx–tag Antibody (Assay Matrix PTY LTD) and secondary IRDye^®^ 800 CW Goat anti-Mouse IgG (Licor, Lincoln, NE, U.S.A.) in TBST buffer (Tris-buffered-saline+ Tween 0.1% w/v) at the suggested dilutions by the manufactures. PageRuler™ Prestained NIR Protein Ladder (Thermofisher, Scoresby, VIC, Australia) was used as a molecular weight ladder.

### 4.7. Fluoresent Labelling of TP

TP containing the introduced cysteine was labelled with Maleimide-Alexaflour 594. The TP was first reduced with TCEP (1 mM) for 30 min. The reducing reaction was followed by spin desalting using 40K–Zeba column (Thermofisher, Scoresby, VIC, Australia) using 20 mM TrisAcetate pH 8.3 and 5% glycerol. The dye which was suspended in pure DMSO and was added to the protein under vortexing so the final concentration of the dye was 500 µM and DMSO was no more than 5% in the final solution. The mixture was incubated for two hours at room temperature and desalted twice with Zeba column as above. The dye was aliquoted and stored at −80 °C.

### 4.8. Cell Culture, Digitonin Permeabilisation Assay (DPA) and Microinjection

Hela cells were grown from a low passage number and were free from mycoplasma contamination after testing. For plating, cells were grown in DMEM supplemented with high glucose, pyruvate and glutamax in addition to 10% FBS (Thermofisher, Scoresby, VIC, Australia). Twenty-five thousand cells were plated onto a (22 × 22 mm) glass coverslip inside a 6-well plate for the digitonin permeabilisation assay (DPA) or in a µ–dish 35 mm (ibidi, Martinsried, Planegg, Germany) for microinjection. After plating, the cells were incubated in 37 °C, 5% CO_2._

For microinjection, the cells were washed with PBS once, and DMEM was replaced with L15, phenol red-free (Thermofisher, Scoresby, VIC, Australia) and incubated in a 37 °C oven without CO_2_ until the protein/dye was ready. To prepare materials for injection, 5µl TP–AF594 was mixed with 5 uL BSA–FITC [29]. The mixture was filtered through 0.22 µM syringe filter (Merck Millipore, Bayswater, VIC, Australia) mounted on a PCR tube which was placed inside a 1.5 mL tube. The mixture was spun for 10 min at 10,000 g. In total, 2 µL of the eluted mixture was then loaded into a microneedle (Eppendorf, Macquarie Park, NSW, Australia) and the cells were injected using Eppendorf InjectMan, with settings of pc = 30 pa, pi = 50 pa, 0.2 sec. Cells were washed with L15/5% FBS once and incubated for ~20 min before imaging. Commercially available L15 media allowed the incubation of cells without the need for CO_2_. Microscopy imaging of microinjection was carried out using Nikon Ti–E (Minato City, Tokyo, Japan) using the wide-field Coolsnap™ camera, and the cells were kept 37 °C during the imaging.

DPA was first optimised to determine the most appropriate digitonin concentration. Digitonin at 20 µg/µL and an incubation period of 5 min on the ice were found to be optimum conditions in our hands. The plate with coverslip-plated cells was moved on the ice and washed twice with cold transport buffer (HEPES 20 mM; potassium acetate 110 mM; sodium acetate 5 mM; magnesium acetate 2 mM; EGTA 1 mM; 1 µg/µL of leupeptin, pepstatin, and aprotinin). Cells were then incubated with 1 mL (20 µg/mL) digitonin in transport buffer for 5 min. This was followed by four washes of cold transport buffer. The coverslips were then mounted facing down on a parafilm sheet with a drop (total 50 µL) of 25 µL 2x import buffer and either 25 µL rabbit reticulocyte lysate (Promega) or 25 µL of 5 mg/mL BSA as a control. Import buffer comprised of transport buffer + energy: 1 mM ATP, 0.2 mM GTP, 5 mM creatine phosphate, 17.5 U/mL creatine phosphokinase. Finally, TP = 1 µM final concentration was used, and 1 µg of anti-DNA mouse antibody (Abcam, Cambridge, United Kingdom) was added to the final 50 µL solution.

Permeabilized cells on a drop were then incubated at 30 °C for 30 min. Cells were washed twice with cold transport buffer and once with cold PBS. Cells were then fixed with 4% paraformaldehyde (Sigma-Aldrich, North Ryde BC NSW, Australia) in PBS for 10 min at room temperature followed by two PBS–BSA 1% washes and labelling with secondary antibody (AF–488; Thermofisher, Scoresby, VIC, Australia) against anti–DNA antibody for two hours in PBS–BSA 1% buffer. Cells were then washed twice with PBS, followed by a third wash for 5 min containing 1X Hoechst counterstain (Thermofisher, Scoresby, VIC, Australia). Nuclear dye was finally washed off with PBS and coverslips were mounted on a glass slide containing a drop of antifade-gold (Thermofisher, Scoresby, VIC, Australia). Ten minutes after mounting, the edges were sealed with nail polish, and slides were either imaged the same day or were imaged the next day.

### 4.9. Imp α/β Protein Expression and Purification

Imp α2 and Imp β1 were expressed as GST–tagged proteins in BL21 (pREP4) *E. coli* cells under native conditions and purified as previously [30]. A GST cleaved Impβ1 protein was also prepared for use in the Impα/β1 heterodimer. Briefly, Impβ1 was bound to GST beads (GE Healthcare, North Richland Hills, TX, USA) and incubated with intracellular buffer (IB, 110 mM KCl, 5 mM NaHCO_3_, 5 mM MgCl_2_, 1 mM EGTA, 0.1 mM CaCl_2,_ 20 mM HEPES, pH 7.4), 400 µM DTT, and 10 U thrombin (Sigma-Aldrich, North Ryde BC NSW, Australia) at room temperature for 7 h. The supernatant was then added to benazamidine sepharose B beads (Pharmacia Biotech, Piscataway, NJ, USA) and incubated overnight at 4 °C. Unbound, cleaved Imp β1 protein was then collected and stored at −80 °C. Before use, the GST cleaved Imp β1, and Imp α2 were dimerised at a 1:1 ratio in IB supplemented with 1 mM DTT at a concentration of 13.6 µM for 15 min at room temperature [31]. Protein concentrations were estimated by Coomassie staining and Bradford Protein Assay (Bio-Rad, Glandesville, NSW, Australia).

### 4.10. AlphaScreen Assay

AlphaScreens were performed in triplicate in opaque 384–well plates in a final volume of 25 µM as previously described [32,33,34]. Briefly, 5 µL of 0.5% BSA was added to wells followed by 5 µL of TP or Trx alone (final concentration, 30 nM) and serial dilutions in 5 µL of the binding partner (importins or GST control). Plates were then incubated for 30 min to allow binding. In total, 5 µL of nickel NTA donor beads (1/250 dilution) was added to wells and incubated for 90 min, then 5 µL of GST acceptor bead (1/250 dilution) was added, and plates were incubated a further 2 h before reading on an Enspire plate reader (PerkinElmer, Waltham, Massachusetts, United States). All dilutions were in PBS, and all incubations were in the dark at room temperature. Curves were plotted using a one-phase association model in Prism (GraphPad Software, Inc., San Diego, CA, USA) to determine the dissociation constants (K_d_) and the maximal signals (B_max_).

### 4.11. Microinjection with Importins and Exportin Inhibitors

Ivermectin, leptomycin B and importazole were all purchased from Sigma. All stocks were dissolved in DMSO. Fifty thousand cells were plated using DMEM/FBS 10% as described in the microinjection section above. The cells were plated in a gridded 33 mm plate (ibidi, Martinsried, Planegg, Germany). Twenty-four hours after plating, new media containing either ivermectin (30 µM), importazole (40 µM), or leptomycin (10 ng/uL) were used with the cells after washing off the older media. Before microinjection, L15 medium containing the same concentration of the inhibitors replaced the DMEM/FBS10% to allow for CO_2_ free imaging. Cells were first microinjected as above using 0.2–0.3 s, 100 pa injection pressure. TP–AF594 in combination with BSA–FITC (no NLS) was used. For positive control, BSA–NLS was mixed with BSA–TexasRed. Both labelled protein solutions were filtered first before loading into the microneedle. Cells were imaged 20 min after microinjection throughout all conditions and were first incubated for 5 min with L15 + Drug + Hoechst counterstain solution (Thermofisher, Scoresby, VIC, Australia). Cells were imaged using Leica SP8 with an HC PL APO CS2 63x/1.4 oil-immersion objective (37 °C). The analysed cells were from at least 2–3 biological repeats on separate days.

### 4.12. Imaging, Data Collection and Analysis

Images were analysed with image-J software using a built-in script to help automate the process of analysing a large number of acquired images. This was achieved by measuring the mean green fluorescence signal in the nucleus (using the blue channel as a nuclear area mask) divided by the mean green fluorescence in the cytoplasm (N_f_/C_f_). N_f_/C_f_ ratios of different fragments were then dot-plotted using GraphPad Prism, and statistical significance was determined using one-way ANOVA followed by Tukey’s multiple comparisons between all conditions within the same cell line. In some instances, Welch’s t–test was also used to compare directly between independent wells. During image acquisition, plated cells were identified, and the data were acquired to cover all transfected cells that we could find. Acquired images were either all analysed or selected randomly before analysis. Nuclear masks were produced automatically using ImageJ software; cell areas were drawn manually around every identified transfected cell using CellMask DeepRed or bright-field images as identifiers. Apoptotic cells and identifiable debris were excluded from the analysis. The acquired images were analysed using a minimum of 15 cells per condition, on average 25 cells per condition (see Appendix A for the number of cells per each condition) and only N_f_/C_f_ < 100 were plotted. An N_f_/C_f_ of 100 and above signifies a complete localisation in the nucleus, and these measurements only existed in fragments where N_f_/C_f_ was very high and had predominant nuclear localisation. The analysed cells were from 2 biological replicates (different days of transfection) using two different cell lines.

For microinjection and DPA, the Hoechst signal was used to create a nuclear mask. AF594 fluorescence intensity signal was measured inside that mask to determine nuclear localisation. Cell membranes were also outlined, and the total fluorescence intensity was measured. Finally, to calculate the cytoplasmic intensity, nuclear fluorescence was subtracted from the total cell fluorescence. Cells with nuclear positive cytoplasmic indicators were excluded as this indicates nuclear microinjection.

Data were plotted as the mean with standard deviation and analysed using either Welch’s t–test (for initial microinjection) or ANOVA with Tukey’s multiple comparisons between the conditions (in case of DPA and microinjections in the presence of inhibitors).

## Figures and Tables

**Figure 1 ijms-22-03310-f001:**
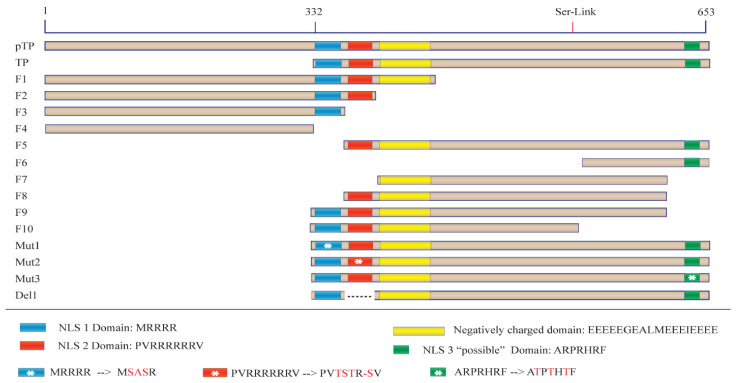
Fragmentation and mutation of preterminal protein (pTP) and terminal protein (TP). Fusion proteins were produced between the C–terminus of GFP and the N–terminus of TP fragments. The constructed fragments are shown with sublegends describing critical domains of interest and mutations are labelled with an asterisk. Ser-link represents the approximate position of the amino acid where TP binds to the viral genome. The starting amino acids of the open reading frame (ORF) is presented as 1 in the figure (see Section 4.1 in Materials and Methods). Mutation details are presented in the legend, and the dashed line signifies a deletion of wild-type amino acids.

**Figure 2 ijms-22-03310-f002:**
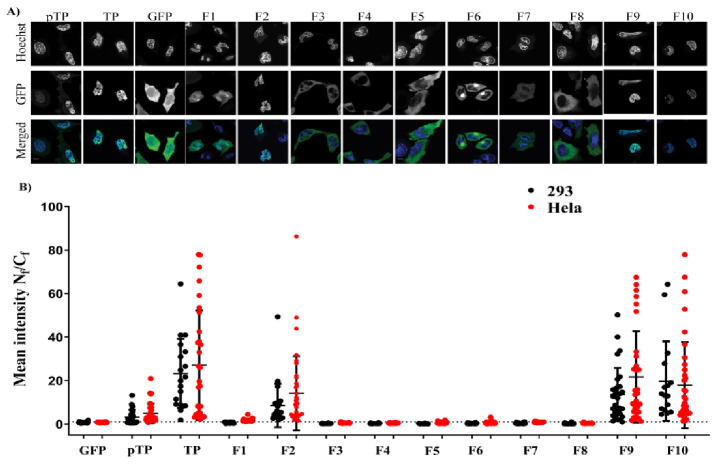
Expression and localisation of pTP, TP, pMax, F1–F10 fragments. Plasmids containing the fragments fused to the C–terminal of GFP or GFP plasmid alone (designated pMax) were transfected into HeLa cells (**A**) Fragments: 1–10, GFP, TP and pTP. (**B**) The mean fluorescence intensity ratio between the nucleus and cytoplasm of the fused fragments. Total fluorescence in the nuclear region (N_f_) and the cytoplasmic region (C_f_) was measured and calculated as described in the Methods section and plotted as N_f_/C_f_ values. Analysis of the graph was performed using one-way ANOVA followed by Tukey’s multiple comparisons analysis (details of the comparisons are described in Appendix A and Appendix A). The dotted line represents an N_f_/C_f_ value of 1. Each dot represents a single cell. Data were pooled from two separate transfections, and all plasmids were sequenced before transfections. The mean values and the exact number of replicates for each condition are described in Appendix A. Each condition was represented by at least 15 cells. All data are represented as mean ± standard deviation. Scale bar = 10 µm.

**Figure 3 ijms-22-03310-f003:**
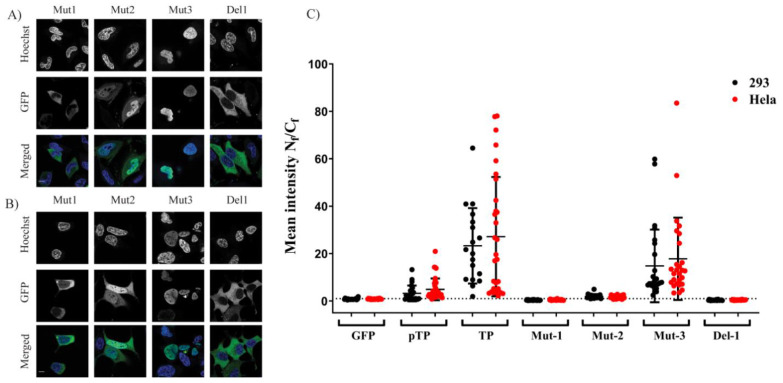
Expression and localisation TP mutants. Specific locations in TP–GFP plasmid were either mutated or deleted as represented in Figure 1. HeLa (**A**) or 293A (**B**) cells were imaged and were presented as detailed in Figure 2 legend. Bar = 10 µM. (**C**) The mean fluorescence intensity ratio between the nucleus and cytoplasm of the mutants. Data were calculated and plotted similar to Figure 2B. Scale bar = 10 µM.

**Figure 4 ijms-22-03310-f004:**
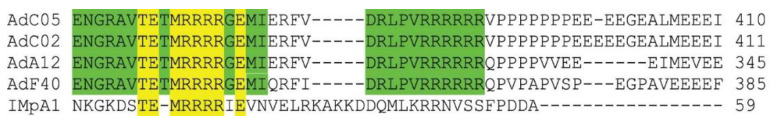
Protein sequence homology of the adenovirus bipartite nuclear localisation signal (NLS) and importin α. Sequence homology of amino acids in the NLS_1_ region of adenovirus TP and importin α is highlighted in yellow. Sequence homologies between different adenovirus serotypes around NLS_1_ and NLS_2_ are highlighted in green. Adenoviruses are represented as “Ad” followed by “C, A, or F” for the species designation and numbers representing the serotypes. Importin α is aligned for comparison. Homology was performed using the Uniprot alignment service.

**Figure 5 ijms-22-03310-f005:**
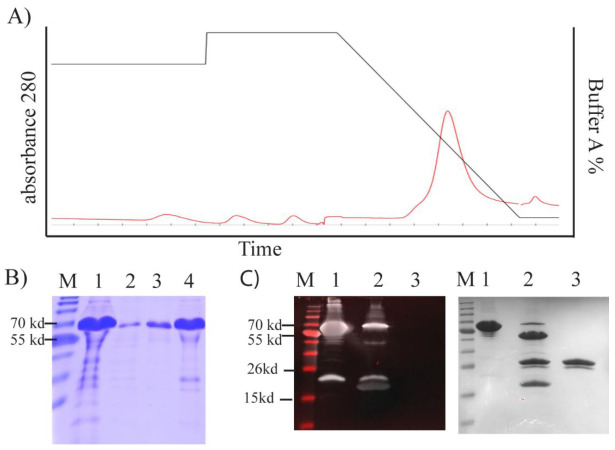
Bacterially expressed TP–Trx or TP purification and characterisation. (**A**) FPLC diagram of TP–Trx elution from Histrap column with decreasing buffer A (i.e., increasing elution buffer B) shown in black and the relative absorbance at λ280 is shown in red. (**B**) SDS–page and Coomassie Blue staining of samples taken before and during TP purification. 1: TP in inclusion bodies; 2: flow-through; 3: wash; 4: elution (pooled). (**C**) Analysis of TP Western blotting. M: NIR ladder; 1: TP–Trx; 2: TPTrx after TEV enzyme cleavage; 3: TEV enzyme only. All wells were run in parallel using the same gel. The gel was cut in half. The first half was stained with Coomassie Blue (right image), and the other half was transferred into a PVDF membrane and stained with the antibody (left image) as described in the Methods section. Antibody staining of the membrane was performed using the Odyssey imaging system and Coomassie staining using Biorad Chemidoc imager.

**Figure 6 ijms-22-03310-f006:**
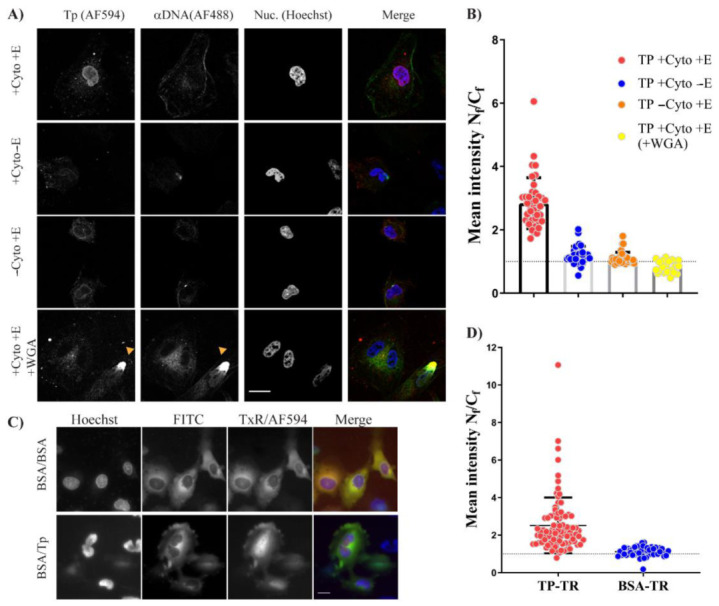
TP–Trx is translocated into the nucleus using active transport that requires energy. (**A**) TP–Trx labelled with AF594 was incubated with cells permeabilised with digitonin for 30 min as described in the Methods section. Images were taken with laser confocal microscopy using the same imaging criteria across all conditions. AF488 (αDNA) was used to determine whether the nuclear membrane is compromised. The orange triangle shows an example of the compromised nucleus next to two cells with an uncompromised nuclear membrane. Those compromised nuclei were excluded from the analysis and calculations. E: energy; Cyto: rabbit reticulocyte lysate cytoplasm; WGA: wheat germ agglutinin. (**B**) Analysis of the mean fluorescence (AF594) intensity ratio (N_f_/C_f_) from DPA experiments. Data collected from fluorescence images in (**A**) were analysed using Image J as described in the Methods section and mean fluorescence intensities were plotted. The dotted line represents N_f_/C_f_ of 1. Difference between conditions was analysed using ANOVA test with Tukey’s multiple comparisons as described in Appendix A. Bars represent mean ± SD. N > 23. (**C**) Cytoplasmic microinjection of labelled TP–Trx–AF594 with BSA–FITC to serve as a cytoplasmic control. As an additional control, BSA–TxR was used instead of TP–Trx–AF594 and coinjected, cytoplasmically, with BSA–FITC. Images were taken approximately 30 min after microinjection of the proteins. (**D**) N_f_/C_f_ ratios between TP–Trx injected compared to BSA control. Analysis of difference between the two conditions was conducted using Welch’s t–test with *p*–value < 0.0001. N > 50, mean ± SD. Scale bar = 10 µM.

**Figure 7 ijms-22-03310-f007:**
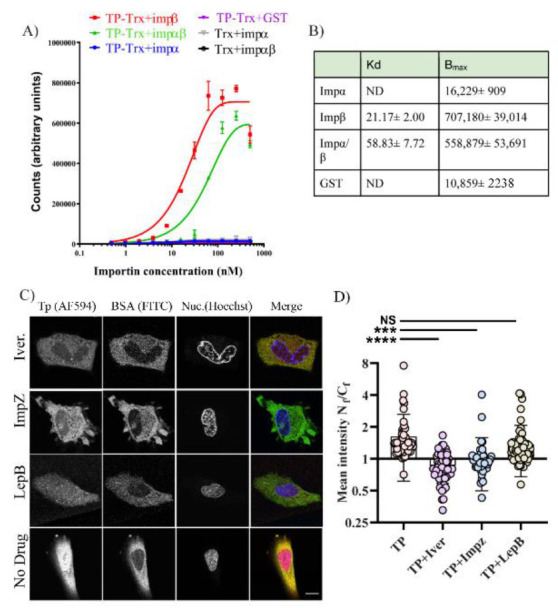
TP–Trx binds to both Impβ1 and the Impα/β1 heterodimer and inhibitors of importins lower TP–Trx nuclear accumulation. (**A**) Triplicate wells containing 30 nM TP were incubated with increasing concentrations of Impα, Impβ1, Impα/β1 heterodimer, or a GST polypeptide control. Single representative curves are shown. As a control, the tags alone without TP were also tested and did not bind with Impα/β1 heterodimer or Impα using 60 nM concentrations of Trx tag. The tag has no binding affinity to both Imps. (**B**) Pooled data from AlphaScreens for binding affinities (K_d_) and maximal binding (B_max_). Data are displayed as mean ± SEM (n = 4). ND, signals were too low to determine K_d_s. (**C**) Representative images from microinjection experiment described and quantified in (**D**). TP–Trx was microinjected using a combination of TP–Trx and BSA–FITC in Hela cells. Before microinjection, the cells were incubated for 3 h with either DMSO, or with ivermectin (30 µM), importazole (40 µM), and leptomycin (10 ng/ul). The microscopy images show a change of TP–Trx distribution using either ivermectin or importazole. Acquiring images started 20–30 min after microinjection. Scale bar = 15 µM. (**D**) Results from calculating TP N_f_/C_f_ after microinjecting TP–Trx with BSA cytoplasmic marker. Cells were incubated with Leptomycin B (LepB), Importazole (ImpZ) or Ivermectin (Iver) or were examined in the absence of the drug. Graphs presented as mean± SD. Data were derived from at least two independent biological replicates (n_cells_ ≥ 45). NS, not significant. *** is *p*–value = 0.0002 and **** is a *p*–value < 0.0001.

## Data Availability

The data presented in this study are available on request from the corresponding author.

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
