# Peer review of "Adenovirus Terminal Protein Contains a Bipartite Nuclear Localisation Signal Essential for Its Import into the Nucleus"

_ijms, 2021, doi:10.3390/ijms22073310_

Round 1

Reviewer 1 Report

The authors describe a detailed characterization of the Adenovirus Terminal Protein (TP) Nuclear Localization Signal (NLS). Using a combination of TP deletions/mutants and a variety of techniques (transfection of plasmids, microinjection of purified proteins, digitonin permeabilization assays and binding assays), they have discovered a previously unrecognized sequence required for TP nuclear import. The main findings are the existence of a bipartite NLS and the involvement of Importin β and Importin α/β heterodimers in the internalization process.

The manuscript is well organized and easy to follow. All conclusions are supported by the experimental data. Although the relevance of these findings on Adenovirus biology will require further work using viral mutants, the results provide strong basic evidences with potential interest for biotechnology applications.

Minor comments:

  1. In figure 4 legend, the “A, C and F” letters in the designation of adenoviruses correspond to species, not strains.
  2. Please revise the text of paragraph 2.3, lines 210 and 211.
  3. Please revise the description of panels in the legend of figure 7.
  4. The authors claim a significant effect of importazole and ivermectin in figure 7, but the result of the statistical analysis is not shown.

Author Response

***Response to Reviewer 1 Comments

1- In figure 4 legend, the “A, C and F” letters in the designation of adenoviruses correspond to species, not strains.

Response 1: Yes, we agree this is an error. The error in the designation is now fixed. The sentence in line 196-197 will now reads as "Adenoviruses are represented as “Ad” followed by “C, A, or F” for the species designation"

         2- Please revise the text of paragraph 2.3, lines 210 and 211.

Response 2: The revision is now made to add clarity to that sentence. The phrase "inclusion bodies protein expression" was added. The full sentence in lines 210 and 211 (214- revised manuscript) now read "Protein expression was confined predominantly to inclusion bodies despite numerous attempts to avoid inclusion bodies protein expression. "

          3- Please revise the description of panels in the legend of figure 7.

Response 3: Agree. line 309 now reads "GST polypeptide control".  In addition, legend (B-D) were rearranged to reflect the arrangement of the figures. (C) in the original draft is now (B). (B) in the original draft is now (D). (D) in the original draft is now (C). The full rearranged text now reads from line 310 reads as " The tag has no binding affinity to both Imps. (B) Pooled data from AlphaScreens for binding affinities (Kd) and maximal binding (Bmax).  Data are displayed as mean ± SEM (n=4).  ND, signals were too low to determine Kds. C) Representative images from microinjection experiment described and quantified in (D). TP-Trx was microinjected using a combination of TP-Trx and BSA-FITC in Hela cells. Before microinjection, the cells were incubated for 3 hours with either DMSO, or with Ivermectin (30µM), Importazole (40µM), leptomycin (10ng/ul). The microscopy images show a change of TP-Trx distribution using either ivermectin or importazole. Acquiring images started 20-30 minutes after microinjection. Scale bar = 15µm.(D) Results from calculating TP Nf/Cf after microinjecting TP-Trx with BSA cytoplasmic marker. Cells were incubated with Leptomycin B (LepB), Importazole (ImpZ) or Ivermectin (Iver) or were examined in the absence of the drug. Graphs presented as mean± SD. Data were derived from at least two independent biological replicates (ncells ≥ 45). "

4- The authors claim a significant effect of importazole and ivermectin in figure 7, but the result of the statistical analysis is not shown.

Response 4: Agree. The description in the results will benefit from referencing to Table S7 and an explicit description of the significance. Now the sentence in line 301 (revised manuscript) will read as "When the data were analysed statistically (Table S7), the Nf/Cf values obtained in the presence of the inhibitors was significantly different from the DMSO control (TP vs. TP+Iver with p-value <0.0001 and; TP vs. TP+Impz p-value =0.0002). "

Line 308 (revised manuscript) In addition, the statistical description was also added to the leptomycin results description "(TP+Impz p-value =0.2024)". The descriptions of the figures now placed at line 304. 

Finally, in figure 7 D, an Indication of significance is placed inside the figure. An updated description in figure 7 legend (point D) now reads "NS, not significant. *** is p-value= 0.0002, **** is a p-value <0.0001." (line 320).

*** track-changes were used in the word file to indicate the changes to indicate changes in the manuscript.

Reviewer 2 Report

This manuscript entitled ‚ Adenovirus terminal protein contains a bipartite nuclear localisation signal essential for its import into the nucleus’ by Hareth A. Al-Wassiti invested the potential NLS motifs within TP using molecular and cellular biology techniques to identify the motifs needed for optimum nuclear import.

Three potential NLS motifs and one negatively charged domain were studied regarding the ability of nuclear entry in transfected cells. In general, the study is well designed; the examination was detailed and complete. However, some issues to be considered:

  1. The whole study is based on GFP-TP fragment fusion protein. GFP fusion protein is nice model to simplify the research procedure; however, the real-adenovirus-word effect is still under question. It would be interesting to bring at least few settings to the adenovirus genome content. If not for the current paper, the author should discuss this point.
  2. Information on how the three potential NLS motifs was identified is missing.
  3. In figure 1, MEHFLP is mentioned in the legend, but not indicated in the figure.
  4. Figure2, in the legend is A and C?
  5. Figure 4, line 194, C, A, or F represent species not strain.

Author Response

***Response to Reviewer 2 Comments

  • The whole study is based on GFP-TP fragment fusion protein. GFP fusion protein is nice model to simplify the research procedure; however, the real-adenovirus-word effect is still under question. It would be interesting to bring at least few settings to the adenovirus genome content. If not for the current paper, the author should discuss this point.

Response 1: We agree. The role of TP on Adenovirus entry is not immediately clear. The role of TP in the adenoviral entry is beyond the aims of our study, which is: to provide a basic understanding of NLSs of TP. The development of viral particles with mutant TPs may not be immediately possible and may require significant and novel assay development. Since TP is involved in the first step of replication and packaging (replication cycle) of adenoviruses, mutations of TP may prevent the generation of viral particles needed to study the phenomenon of viral DNA entry (infectious cycle). We added a brief discussion on those points: Line 403: "

"The current study focuses on identifying and studying the nuclear localisation sequences of TP and its biochemical interactions with host importins. This aim provides a basic and comprehensive understanding of the mechanism of entry of this protein, prior to studying the TP in the context of viral entry. Future work involving the study of viral particles with mutant TP will be important to clarify the rule of TP in viral entry. However, studying adenoviral TP in the context of nuclear pore entry cycle may pose certain technical challenges. Nuclear localisation of newly expressed TP is a vital first step of adenovirus replication and generating viral particles. Mutations to prevent entry can stop the generation of reporter viral particles for studying the phenomenon of viral nuclear entry. Therefore, technique development that circumvents such challenge will be needed."

  • Information on how the three potential NLS motifs was identified is missing.

Response 2: In line 93, we described that "Inspection of the pTP sequence suggested three sequences with positively charged amino acids, hereafter referred to as NLS1, NLS2 and NLS3. "

The suggestion is based on the rich positively charged motifs. Our hypothesis is based on classical NLS description where NLS mono and bipartite are rich in basic amino acids residues. (described in :EMBO Rep. 2000 Nov 15; 1(5): 411–415). However, we realise that the description of NLS is not exhaustive. Therefore, in addition to the mutation of basic amino acids, we also generated 10 fragments to investigate if there was any other NLS outside our hypothesised positively charged NLSs.

Line 101. A description was added to assist the clarity of the source: 

"This hypothesis is based on identifying motifs with rich basic residues that sometimes correlates with the presence of NLS. However, we performed comprehensive fragmentation analysis beyond the hypothesized NLSs. "

  • In figure 1, MEHFLP is mentioned in the legend, but not indicated in the figure.

Response 3: MEHFLP represents the starting point of TP. This is described in the materials and method (from Gene bank accession AAA92208.1)

The sentence in the legend is now reworded for clarity to reflect the starting point and references the methods section.

Line 97: "The starting amino acids of the ORF is presented as 1 in the figure (see section 4.1 in Materials and Methods)"

  • Figure2, in the legend, is A and C?

Response 4: Yes, the error is now corrected from legend (C) to "(B)".

  • Figure 4, line 194, C, A, or F represent species not strain.

Response 5: Yes, we agree. The error in the designation is now fixed. The sentence in line 196-197 will now reads as "Adenoviruses are represented as “Ad” followed by “C, A, or F” for the species designation"

*** track-changes were used in the word file to indicate the changes to indicate changes in the manuscript.

Reviewer 3 Report

In their manuscript „Adenovirus terminal protein contains a bipartite nuclear localization signal essential for its import into the nucleus” As-Wassiti at al., provide a detailed in-depth analyses the adenoviral terminal protein (TP) regarding its nuclear localization signals and nuclear import. The authors constructed several deletion and mutated version of TP and analyzed its import behavior using several biochemical and bioengineering methods. The manuscript is well structured and methods as well as results are presented in a clear and comprehensive way.

The results obtained in this study will contribute to our understanding of nuclear import in general and of intracellular trafficking of adenovirus particles to the nucleus in particular. However, to open the scope of this manuscript to a broader audience, I strongly encourage the authors to address the points listed below.

  1. Introduction, lines 45-48: Authors state that after translation, pTP is translocated in the nucleus and cleaved later on during particle maturation. Therefore, one would assume that expressed pTP is efficiently imported into the nucleus. However, results shown in Figure 2 and 3 indicate that the nuclear import of pTP (compared to TP) is rather incomplete. Can you please comment.
  2. Results section, lines 107-111. Although transparently explained, the decision to use a Students t-test instead of one-way ANOVA is not fully comprehensible. I therefore suggest to mitigate this sentence, e.g. “…testing different transgenes, we suggest to apply a direct statistical analysis using t-test…”.
  3. Since the results obtained by Al-Wassiti et al may be of interested for a wider audience, I strongly encourage the authors to discuss their findings in the context of our current knowledge of adenoviral cytosolic trafficking, attachment to the nuclear pore complex and beyond.
  4. Further, these findings may provide a base for the development of anti-viral strategies. This should be briefly discussed as well.
  5. Caption Figure 2, line 124: It should be (B) instead of (C).
  6. Results section, line 212-213: a closing bracket is missing
  7. Figure 7C: Is should be Hoechst instead of Hoescht.

Author Response

Response to Reviewer 3 Comments

  • Introduction, lines 45-48: Authors state that after translation, pTP is translocated in the nucleus and cleaved later on during particle maturation. Therefore, one would assume that expressed pTP is efficiently imported into the nucleus. However, results shown in Figure 2 and 3 indicate that the nuclear import of pTP (compared to TP) is rather incomplete. Can you please comment.

Response 1: pTP initiates viral replication of the incoming viral DNA and is then cleaved by viral proteases to ultimately generates TP (Which is covalently bound to DNA). Importantly, this cleavage is generated by viral protease (A. WEBSTER, JOURNAL OF VIROLOGY, Nov. 1994, p. 7292-7300). In our study, pTP is predominately located in the nucleus. With N/C ratios of (3.19 and 4.91) for 293 and Hela indicating higher accumulation in the nucleus compared to cytoplasm.  Microscopy images of pTP in Hela and 293 also show a pronounced localisation in the nucleus. Indeed, the mean N/C of TP was higher as indicated in figure 2. This can be attributed to a few potential issues:  

  • Presence of structural constraints that may slow pTP diffusion into the nucleus. However, this can be difficult to study without structural information on pTP.
  • We can't exclude the possibility of the presence of a nuclear export signal in the pTP sequence. Although we can't identify such a sequence. Neither it is within the scope to study nuclear export.
  • Finally, the size of pTP is the largest of all fragments (combined with the size of GFP). While the nuclear pore can allow large proteins to entre the nucleus, smaller fragments in our study with both intact NLS1 and NLS2 were localised in a higher ratio into the nucleus. 

  1. Results section, lines 107-111. Although transparently explained, the decision to use a Students t-test instead of one-way ANOVA is not fully comprehensible. I therefore suggest to mitigate this sentence, e.g. “…testing different transgenes, we suggest to apply a direct statistical analysis using t-test…”.

Response 2: The description is now restated into this form:

line 109 "Statistical analysis showed that in HeLa cells Nf/Cf of GFP was significantly different to that of GFP-TP (p-value <0.0001) but not GFP-pTP using Tukey post-hoc analysis (p-value 0.9954; see also Table S4 and Table S5). " 

The following sentences were removed to avoid any potential confusion in using the student t-test.

" However, since these are independent wells testing different transgenes, we also applied direct statistical analysis using t-test without assuming similar standard deviations (Welch test). In this context, Nf/Cf pTP was statistically significant from GFP in both cell lines (p= 0.0024 for 293A cell line and p<0.0001 for Hela cell comparisons). "

  • Since the results obtained by Al-Wassiti et al may be of interested for a wider audience, I strongly encourage the authors to discuss their findings in the context of our current knowledge of adenoviral cytosolic trafficking, attachment to the nuclear pore complex and beyond.

Response 3: a brief description was added to discuss the context of trafficking within the scope of Adenviral docking and nuclear translocation Line 419. "Our work proposes a new question. Is TP involved in adenoviral trafficking to the nucleus? Intracellular viral trafficking is thought to be carried out by hexon where the virus uses to dock into the nuclear pore [2]. However, this docking may be incomplete [1]. Previous work showed some involvement of the condensing protein, pVII, in viral delivery across the nucleus [7]. However, in that study, TP was also present in the incoming viral DNA. Tp is present only as two protein copies per viral particle, which makes its identification and localisation difficult. Recently, a biotechnological approach to use TP-DNA as a cloning vector helped produce a higher titre of adenovirus after transfection into permissive cell lines [34]."  

Other discussion sections were added as requested by other reviewers that discuss other aspects of Tp involvement in the viral cycle.

Reference [34] was also added to the bibliography.

  • Further, these findings may provide a base for the development of anti-viral strategies. This should be briefly discussed as well.

Response 4: Brief discussion on the potential intervention with TP trafficking is now added. Line 392 (revised manuscript).

"Our study offers an opportunity to study antiviral interventions. Indeed, TP entry to the nucleus is crucial for viral replication and infection [5]. Our study also highlights the possibility of TP involvement in incoming viral DNA entry. Intervening with TP entry to the nucleus can be useful to mitigate adenoviral infections. Adenoviruses are common viruses with respiratory, neurological, eye and gastrointestinal infections (33). In our study, the use of ivermectin and importazol helped reduce TP entry to the nucleus. Both these drugs and others can be tested on adenoviruses and the impact of drugs on viral entry and replication. Our work shows intimate and high-affinity binding between Importin αβ or Importin β. But structural studies can be used to probe whether TP: Importins interaction is unique or whether it can be disrupted by other drugs that are known to affect importin trafficking or interaction."

Reference [33] was also added to the bibliography.

  • Caption Figure 2, line 124: It should be (B) instead of (C). 

Response5: Yes, the error is now corrected from legend (C) in line 126 (revised manuscript file) to "(B)".

  • Results section, line 212-213: a closing bracket is missing

Response 6: The revision is now made to add clarity to that sentence. The phrase "inclusion bodies protein expression" was added. The full sentence in lines 213 and 214 now read "Protein expression was confined predominantly to inclusion bodies despite numerous attempts to avoid inclusion bodies protein expression. " The bracket was removed.

  • Figure 7C: Is should be Hoechst instead of Hoescht.

Response 7: The description in Figure 7C is now updated to Hoechst.